# Classification Accuracy Score for Conditional Generative Models

**Suman Ravuri & Oriol Vinyals**[*]
DeepMind
London, UK N1C 4AG
`ravuris, vinyals@google.com`

## Abstract

Deep generative models (DGMs) of images are now sufficiently mature that they produce nearly photorealistic samples and obtain scores similar to the data distribution on heuristics such as Frechet Inception Distance (FID). These results, especially on large-scale datasets such as ImageNet, suggest that DGMs are learning the data distribution in a perceptually meaningful space and can be used in downstream tasks. To test this latter hypothesis, we use class-conditional generative models from a number of model classes—variational autoencoders, autoregressive models, and generative adversarial networks (GANs)—to infer the class labels of real data. We perform this inference by training an image classifier using only synthetic data and using the classifier to predict labels on real data. The performance on this task, which we call *Classification Accuracy Score* (CAS), reveals some surprising results not identified by traditional metrics and constitute our contributions. First, when using a state-of-the-art GAN (BigGAN-deep), Top-1 and Top-5 accuracy decrease by 27.9% and 41.6%, respectively, compared to the original data; and conditional generative models from other model classes, such as Vector-Quantized Variational Autoencoder-2 (VQ-VAE-2) and Hierarchical Autoregressive Models (HAMs), substantially outperform GANs on this benchmark. Second, CAS automatically surfaces particular classes for which generative models failed to capture the data distribution, and were previously unknown in the literature. Third, we find traditional GAN metrics such as Inception Score (IS) and FID neither predictive of CAS nor useful when evaluating non-GAN models. Furthermore, in order to facilitate better diagnoses of generative models, we open-source the proposed metric.

## 1   Introduction

Evaluating generative models of high-dimensional data remains an open problem. Despite a number of subtleties in generative model assessment [1], in a quest to improve generative models of images, researchers, and particularly those who have focused on Generative Adversarial Networks [2], have identified desirable properties such as "sample quality" and "diversity" and proposed automatic metrics to measure these desiderata. As a result, recent years have witnessed a rapid improvement in the quality of deep generative models. While ultimately the utility of these models is their performance in downstream tasks, the focus on these metrics has led to models whose samplers now generate nearly photorealistic images [3–5]. For one model in particular, BigGAN-deep [3], results on standard GAN metrics such as Inception Score (IS) [6] and Frechet Inception Distance (FID) [7] approach those of the data distribution. The results on FID, which purports to be the Wasserstein-2 metric in a perceptual feature space, in particular suggest that BigGANs are capturing the data distribution.

---

[*]Corresponding author: Suman Ravuri (`ravuris@google.com`).

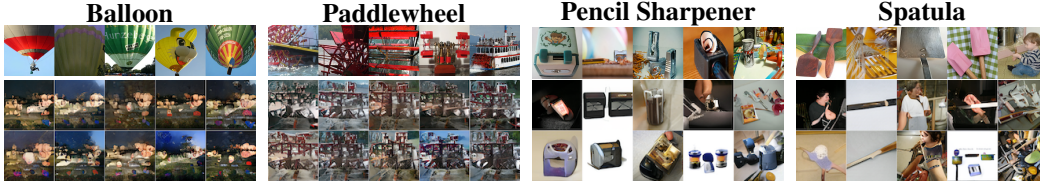

Figure 1: CAS identifies classes for which BigGAN-deep fails to capture the data distribution. Top row are real images, and the bottom two rows are samples from BigGAN-deep.

A similar, though less heralded, improvement has occurred for models whose objectives are (bounds of) likelihood, with the result that many of these models now also produce photorealistic samples. Examples include: Subscale Pixel Networks [8], unconditional autoregressive models of 128×128 ImageNet that achieve state-of-the-art test set log-likelihoods; Hierarchical Autoregressive Models (HAMs) [9], class-conditional autoregressive models of 128×128 and 256×256 ImageNet; and the recently introduced Vector-Quantized Variational Autoencoder-2 (VQ-VAE-2) [10], a variational autoencoder that uses vector quantization and an autoregressive prior to produce high-quality samples. Notably, these models measure diversity using test set likelihood and assess sample quality through visual inspection, eschewing the metrics typically used in GAN research.

As these models increasingly seem "to learn the distribution" according to these metrics, it is natural to consider their use in downstream tasks. Such a view certainly has a precedent: improved test set likelihoods in language models, unconditional models of text, also improve performance in tasks such as speech recognition [11]. While a generative model need not learn the data distribution to perform well on a downstream task, poor performance on such tasks allows us to diagnose specific problems with both our generative models and the task-agnostic metrics we use to evaluate them. To that end, we use a general framework (first posed in [12] and further studied in [13]) in which we use conditional generative models to perform approximate inference and measure the quality of that inference. The idea is simple: for any generative model of the form $p_\theta(x|y)$, we learn an inference network $\hat{p}(y|x)$ using only samples from the conditional generative model and measure the performance of the inference network on a downstream task. We then compare performance to that of an inference network trained on real data.

We apply this framework to conditional image models where $y$ is the image label, $x$ is the image, and task is image classification. (N.B. this approach has been used for evaluating smaller scale GANs [12–16]). The performance measure we use, Top-1 and Top-5 accuracy, denote a *Classification Accuracy Score* (CAS). The gap in performance between networks trained on real and synthetic data allows us to understand specific deficiencies in the generative model. Although a simple metric, CAS reveals some surprising results:

- When using a state-of-the-art GAN (BigGAN-deep) and an off-the-shelf ResNet-50 classifier as the inference network, we found that Top-1 and Top-5 accuracies decrease by 27.9% and 41.6%, respectively, compared to using real data.

- Conditional generative models based on likelihood, such as VQ-VAE-2 and HAM, perform well compared to BigGAN-deep, despite achieving relatively poor Inception Scores and Frechet Inception Distances. Since these models produce visually appealing samples, the result suggests that IS and FID are poor measures of non-GAN model performance.

- CAS automatically surfaces particular classes for which BigGAN-deep and VQ-VAE-2 fail to capture the data distribution and were previously unknown in the literature. Figure 1 shows four such classes for BigGAN-deep.

- We find that neither IS, nor FID, nor combinations thereof are predictive of CAS. As generative models may soon be deployed in downstream tasks, these results suggest that we should create metrics that better measure task performance.

- We calculate a *Naive Augmentation Score* (NAS), a variant of CAS where the image classifier is trained on both real and synthetic images, to demonstrate that classification performance improves in limited circumstances. Augmenting the ImageNet training set with low-diversity BigGAN-deep images improves Top-5 accuracy by 0.2%, while augmenting the dataset with any other synthetic images degrades classification performance.

In Section 2 we provide a few definitions, desiderata of metrics, and shortcomings of the most popular metrics in relation to different research directions for generative modeling. Section 3 defines CAS. Finally, Section 4 provides a large-scale study of current state-of-the-art generative models using FID, IS, and CAS on both the ImageNet and CIFAR-10 datasets.

## 2  Metrics for Generative Models

Much of the difficulty in evaluating any generative model is not knowing the task for which the model will be used. Understanding how the model will be deployed, however, has important implications on its desired properties. For example, consider the seemingly similar tasks of automatic speech recognition and speech synthesis. While both tasks may share the same generative model of speech—such as a hidden Markov Model $p_\theta(o, l)$ with the observed and latent variables being the waveform $o$ and word sequence $l$, respectively—the implications of model misspecification are vastly different. In speech recognition, the model should be able to infer words for all possible speech waveforms, even if the waveforms themselves are degraded. In speech synthesis, however, the model should produce the most realistic-sounding samples, even if it cannot produce all possible speech waveforms. In particular, for automatic speech recognition, we care about $p_\theta(l|o)$, while for speech synthesis, we care about $o \sim p_\theta(o|l)$.

In absence of a known downstream task, we assess to what extent the model distribution $p_\theta(x)$ matches the data distribution $p_{data}(x)$, a less specific and often more difficult goal. Two consequences of the trivial observation that $p_\theta(x) = p_{data}(x)$ are: 1) each sample $x \sim p_\theta(x)$ "comes" from the data distribution (i.e., it is a "plausible" sample from the data distribution), and 2) that all possible examples from the data distribution are represented by the model. Different metrics that evaluate the degree of model mismatch weigh these criteria differently.

Furthermore, we expect our metrics to be relatively fast to calculate. This last desideratum often depends on the model class. The most popular seem to be:

- (Inexact) Likelihood models using variational inference (e.g., VAE [17, 18])
- Likelihood using autoregressive models (e.g., PixelCNN [19])
- Likelihood models based on bijections (e.g., GLOW [20], rNVP [21])
- (Possibly inexact) likelihood using energy-based models (e.g., RBM [22])
- Implicit generative models (e.g., GANs)

For the first four of these classes, the likelihood objective provides us scaled estimates of the KL-divergence between the data and model. Furthermore, test set likelihood is also an implicit measure of diversity. The likelihood, however, is a fairly poor measure of sample quality [1] and often scores out-of-domain data more highly than in-domain data [23].

For implicit models, the objective provides neither an accurate estimate of a statistical divergence or distance nor a natural evaluation metric. The lack of any such metrics likely forced researchers to propose heuristics that measure versions of both 1 and 2 (sample quality and diversity) simultaneously. Inception Score (IS) [6] ($\exp(\mathbb{E}_x[p(y|x)\|p(y)])$) measures 1 by how confidently a classifier assigns an image to a particular class ($p(y|x)$), and 2 by penalizing if too many images were classified to the same class ($p(y)$). More principled versions of this procedure are Frechet Inception Distance (FID) [7] and Kernel Inception Distance (KID) [24], which both use variants of two-sample tests in a learned "perceptual" feature space, the Inception pool3 space, to assess distribution matching. Even though this space was an ad-hoc proposition, recent work [25] suggests that deep features correlate with human perception of similarity. Even more recent work [26, 27] calculate 1 and 2 independently by calculating precision and recall.

Reliance on IS and FID in particular has led to improvement in GAN models but has certain deficiencies. IS does not penalize a lack of intra-class diversity, and certain out-of-distribution samples produce Inception Scores three times higher than that of the data [28]. FID, on the other hand, suffers from a high degree of bias [24]. Moreover, the pool3 feature layer may not even correlate well with human judgment of sample quality [29]. In this work, we also find that non-GAN models have rather poor Inception Scores and Frechet Inception Distances, even though the samples are visually appealing.

Rather than creating ad-hoc heuristics aimed at broadly measuring sample quality and diversity, we instead evaluate generative models by assessing their performance on a downstream task. This is akin to measuring a generative model of speech by evaluating it on automatic speech recognition. Since models considered here are implicit or do not admit exact likelihoods, exact inference is difficult. To circumvent this issue, we train an inference network on samples from the model. If the generative model is indeed capturing the data distribution, then we could replace the original distribution with a model-generated one, perform any downstream task, and obtain the same result. In this work, we study perhaps the simplest downstream task: image classification.

This idea is not necessarily new: for GAN evaluation, it has been independently discovered at least four times. [12] first introduced the metric (denoted "adversarial accuracy") to measure their proposed Layer-Recursive GAN and connected image classification to approximate inference. [13] more systematically studied this idea of approximate inference to measure the boundary distortion induced by GANs. They did this by training separate per-label unconditional generative models, and then trained classifiers on synthetic data to understand how the boundary shifted and to measure the sample diversity of GANs. Predating [13], [14] used "Train on Synthetic, Test on Real" to measure a recurrent conditional GAN for medical data. [16] trained on synthetic data, tested on real (denoted "GAN-train") as an approximate recall metric for GANs. They also trained on real data and tested on synthetic (denoted "GAN-test") as an approximate precision test. Unlike previous work, they tested on larger datasets such as $128 \times 128$ ImageNet, but with smaller scale models such as SNGAN [30].

The metrics mentioned above are by no means the only ones, and researchers have proposed methods to evaluate other properties of generative models. [31] constructs approximate manifolds from data and samples, and applies the method to GAN samples to determine whether mode collapse occurred. [32] attempts to determine the support size of GANs by using a Birthday Paradox test, though the procedure requires a human to identify two nearly-identical samples. Maximum Mean Discrepancy [33] is a two-sample test that has many nice theoretical properties but seems to be less used because the choice of kernels do not necessarily coincide with human judgment. Procedurally similar to our method, [34] proposes a "reverse LM score", which trains a language model on GAN data and tests on a real held-out set. [35] measures the quality of generative models of text by training a sentiment analysis classifier. Finally, [36] classifies real data using a student network mimicking a teacher network pretrained on real data but distilled on GAN data.

Our work most closely mirrors [16], but differs in a some key respects. First, since we view image classification as approximate inference, we are able to describe its limitations in Section 3, and verify the approximation in Section 4.5. Second, while in [16] performance on GAN-train correlates with improved IS and FID, we focus more on large-scale and non-GAN models, such as VQ-VAE-2 and HAMs, where FID and IS are not indicative of classification performance. Third, by polling the inference network, we can identify classes for which the model failed to capture the data distribution. Finally, we open-source the metric for ImageNet for ease of evaluating large-scale generative models.

## 3 Classification Accuracy Score

At the heart of CAS lies a very simple idea: if the model captures the data distribution, performance on any downstream task should be similar whether using the original or model data. To make this intuition more precise, suppose that data comes from a distribution $p(x, y)$, the task is to infer $y$ from $x$, and we suffer a loss $L(y, \hat{y})$ for predicting $\hat{y}$ when the true label is $y$. The *risk* associated with a classifier $\hat{y} = f(x)$ is:

$$\mathbb{E}_{p(x,y)}[L(y, \hat{y})] = \mathbb{E}_{p(x)}[\mathbb{E}_{p(y|x)}[L(y, \hat{y})|X]] \tag{1}$$

As we only have samples from $p(x, y)$, we measure the empirical risk $\frac{1}{N} L(y_i, f(x_i))$. From the right hand side of Equation 1, of the set of predictions $\mathcal{Y}$, the optimal one $\hat{y}$ minimizes the expected posterior loss:

$$\hat{y} = \arg \min_{y' \in \mathcal{Y}} \mathbb{E}_{p(y|x)}[L(y, y')|X] \tag{2}$$

Assuming we know the label distribution $p(y)$, a generative modeling approach to this problem is to model the conditional distribution $p_\theta(x|y)$, and infer labels using Bayes rule: $p_\theta(y|x) = \frac{p_\theta(x|y)p(y)}{p_\theta(x)}$. If $p_\theta(y|x) = p(y|x)$, then we can make predictions that minimize the risk for any loss function. If the risk is not minimized, however, then we can conclude that distributions are not matched, and we can interrogate $p_\theta(y|x)$ to better understand how our generative models failed.

For most modern deep generative models, however, we have access to neither $p_\theta(x|y)$, the probability of the data given the label, nor $p_\theta(y|x)$, the model conditional distribution, nor $p(y|x)$, the true conditional distribution. Instead, from samples $x, y \sim p(y)p_\theta(x|y)$, we train a discriminative model $\hat{p}(y|x)$ to learn $p_\theta(y|x)$, and use it to estimate the expected posterior loss $\mathbb{E}_{\hat{p}(y|x)}[L(y, \hat{y})|X]$. We define the *generative risk* as $\mathbb{E}_{p(x,y)}[L(y, \hat{y}_g)]$, where $\hat{y}_g$ is the classifier that minimizes the expected posterior loss under $\hat{p}(y|x)$. Then we compare the performance of the classifier to the performance of the classifier trained on samples from $p(x, y)$.

In the case of conditional generative models of images, $y$ is the class label for image $x$, and the model of $\hat{p}(y|x)$ is an image classifier. We use ResNets [37] in this work. The loss functions $L$ we explore are the standard ones for image classification. One is 0-1, which yields Top-1 accuracy, and the other is 0-1 in the Top-5, which yields Top-5 accuracy.[2] Procedurally, we train a classifier on synthetic data, and evaluate the performance of the classifier on real data. We call the accuracy the *Classification Accuracy Score* (CAS).

Note that a CAS close to that for the data does not imply that the generative model accurately modeled the data distribution. This may happen for a few reasons. First, $p_\theta(y|x) = p(y|x)$ for any generative model that satisfies $\frac{p_\theta(x|y)}{p_\theta(x)} = \frac{p(x|y)}{p(x)}$ for all $x, y \sim p(x, y)$. One example is a generative model that samples from the true distribution with probability $p$, and from a noise distribution with a support disjoint from the true distribution with probability $1 - p$. In this case, our inference model is good but the underlying generative model is poor.

Second, since the losses considered here are not proper scoring rules [38], one could obtain reasonable CAS from suboptimal inference networks. For example, suppose that $p(y|x) = 1.0$ for the correct class while $\hat{p}(y|x) = 0.51$ for the correct class due to poor synthetic data. CAS for both is 100%. Using a proper scoring rule, such as Brier Score, eliminates this issue, but experimentally we found limited practical benefit from using one.

Finally, a generative model that memorizes the training set will achieve the same CAS as the original data.[3] In general, however, we hope that generative models produce samples disjoint from the set on which they are trained. If the samples are sufficiently different, we can train a classifier on both the original data and model data and expect improved accuracy. We denote the performance of classifiers trained on this "naive augmentation" *Naive Augmentation Score* (NAS). Our CAS results, however, indicate that the current models still significantly underfit, rendering the conclusions less compelling. For completeness, we include results on augmentation in Section 4.4.

Despite these theoretical issues, we find that generative models have Classification Accuracy Scores lower than the original data, indicating that they fail to capture the data distribution.

## 3.1 Computation and Open-Sourcing Metric

Computationally, training classifiers is significantly more demanding than calculating FID or IS over 50,000 samples. We believe, however, that now is the right time for such a metric due to a few key advances in training classifiers: 1) the training of ImageNet classifiers has been reduced to minutes [39], 2) with cloud services, the variance due to implementation details of such a metric is largely mitigated, and 3) the price and time cost of training classifiers on cloud services is reasonable and will only improve over time. Moreover, many class-conditional generative models are computationally expensive to train, and thus, even a relatively expensive metric such as CAS comprises a small percentage of the training budget.

We open-source our metric on Google Cloud for others to use. The instructions are given in Appendix B. At the time of writing, one can compute the metric in 10 hours for roughly \$15, or in 45 minutes for roughly \$85 using TPUs. Moreover, depending on affiliation, one may be able to access TPUs for free using the Tensorflow Research Cloud (TFRC) (https://www.tensorflow.org/tfrc/).

Table 1: CAS for different models at 128×128 and 256×256 resolutions. BigGAN-deep samples are taken from best truncation parameter of 1.5.

| Training Set | Resolution | Top-5 Accuracy | Top-1 Accuracy | IS | FID-50K |
|---|---|---|---|---|---|
| Real | 128×128 | 88.79% | 68.82% | $165.38 \pm 2.84$ | 1.61 |
| BigGAN-deep | 128×128 | 64.44% | 40.64% | $71.31 \pm 1.57$ | 4.22 |
| HAM | 128×128 | **77.33%** | **54.05%** | $17.02 \pm 0.79$ | 46.05 |
| Real | 256×256 | 91.47% | 73.09% | $331.83 \pm 5.00$ | 2.47 |
| BigGAN-deep | 256×256 | 65.92% | 42.65% | $109.39 \pm 1.56$ | 11.78 |
| VQ-VAE-2 | 256×256 | **77.59%** | **54.83%** | $43.44 \pm 0.87$ | 38.05 |

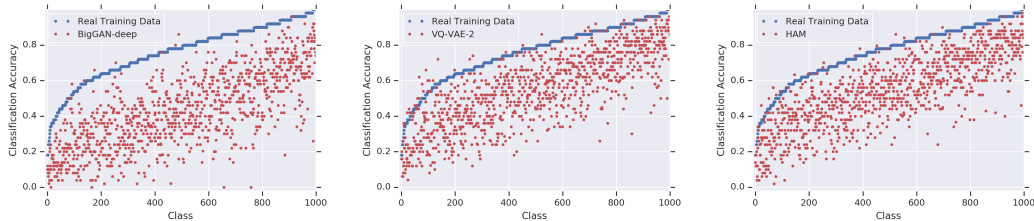

Figure 2: Comparison of per-class accuracy of data (blue) vs. model (red). Left: BigGAN-deep 256×256 at 1.5 truncation level. Middle: VQ-VAE-2 256×256. Right: HAM 128×128.

# 4 Experiments

Our experiments are simple: on ImageNet, we use three generative models—BigGAN-deep at 256×256 and 128×128 resolutions, HAM with masked self-prediction auxiliary decoder at 128×128 resolution, and VQ-VAE-2 at 256×256 resolution—to *replace* the ImageNet training set with a model-generated one, train an image classifier, and evaluate performance on the ImageNet validation set. To calculate CAS, we replace the ImageNet training set with one sampled from the model, and each example from the original training set is replaced with a model sample from the same class.

In addition, we compare CAS to two traditional GAN metrics: IS and FID, as these metrics are the current gold standard for GAN comparison and are being used to compare non-GAN to GAN models. Both rely on a feature space from a classifier trained on ImageNet, suggesting that if metrics are useful at predicting performance on a downstream task, it would indeed be this one.

Further details about the experiment can be found in Appendix A.1.

## 4.1 Model Comparison on ImageNet

Table 1 shows the performance of classifiers trained on model-generated datasets compared to those on the real dataset for 256×256 and 128×128, respectively. At 256×256 resolution, BigGAN-deep achieves a CAS Top-5 of 65.92%, suggesting that BigGANs are learning nontrivial distributions. Perhaps surprisingly, VQ-VAE-2, though performing quite poorly compared to BigGAN-deep on both FID and IS, obtains a CAS Top-5 accuracy of 77.59%. Both models, however, lag the original 256×256 dataset, which achieves a CAS Top-5 Accuracy of 91.47%.

We find nearly identical results for the 128×128 models. BigGAN-deep achieves CAS Top-5 and Top-1 similar to the 256×256 model (note that IS and FID results for 128×128 and 256×256 BigGAN-deep are vastly different). HAMs, similar to VQ-VAE-2, perform poorly on FID and Inception Score but outperform BigGAN-deep on CAS. Moreover, both models underperform relative to the original 128×128 dataset.

## 4.2 Uncovering Model Deficiencies

To better understand what accounts for the gap between generative model and dataset CAS, we broke down the performance by class (Figure 2). As shown in the left pane, nearly every class of BigGAN-deep suffers a drop in performance compared to the original dataset, though six classes—

**BigGAN-deep**　　　　**VQ-VAE-2**　　　　**HAM**

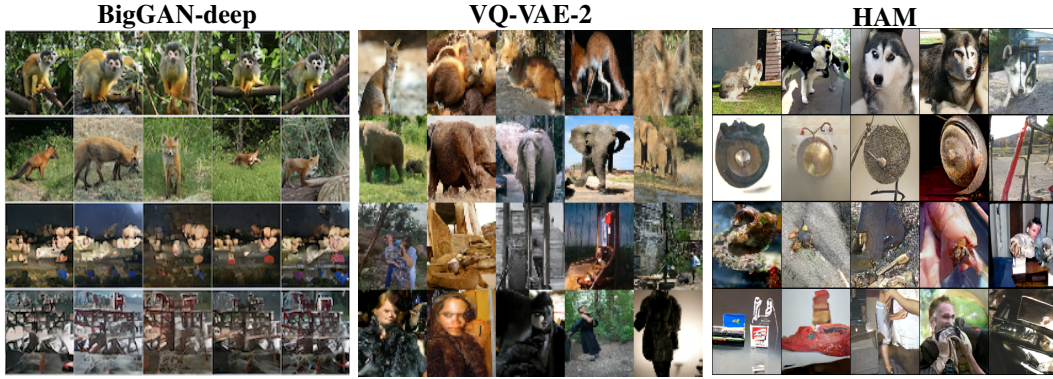

Figure 3: The top two rows are samples from classes that achieved the best test set performance relative to original dataset. The bottom two rows are those from classes that achieved the worst. Left: BigGAN-deep top two—squirrel monkey and red fox—and bottom two—(hot air) balloon and paddlewheel. Middle: VQ-VAE-2 top two—red fox and African elephant—and bottom two—guillotine and fur coat. Right: HAM top two—husky and gong/tim-tam—and bottom two—hermit crab and harmonica.

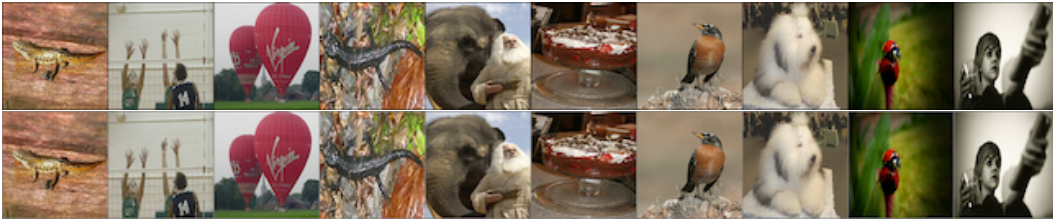

Figure 4: Top: Originals, Bottom: Reconstructions using VQ-VAE-2.

partridge, red fox, jaguar/panther, squirrel monkey, African elephant, and strawberry—show marginal improvement over the original dataset. The left pane of Figure 3 shows the two best and two worst performing categories, as measured by the difference in classification performance. Notably, for the two worst performing categories and two others—balloon, paddlewheel, pencil sharpener, and spatula—classification accuracy was 0% on the validation set.

The per-class breakdown of VQ-VAE-2 (middle pane of Figure 2) shows that this model also underperforms the real data for most classes (only 31 classes perform better than the original data), though the gap is not as large as for BigGAN-deep. Furthermore, VQ-VAE-2 has better generalization performance in 87.6% of classes compared to BigGAN-deep, and suffers 0% classification accuracy for no classes. The middle pane of Figure 3 shows the two best and two worst performing categories.

The right panes of Figures 2 and 3 show the per-class breakdown and top and bottom two classes, respectively, for HAMs. Results broadly mirror those of VQ-VAE-2.

### 4.3 A Note on FID and a Second Note on IS

We note that IS and FID have very little correlation with CAS, suggesting that alternative metrics are needed if we intend to deploy our models on downstream tasks. As a controlled experiment, we calculate CAS, IS, and FID for BigGAN-deep models with input noise distributions truncated at different values (known as the "truncation trick"). As noted in [3], lower truncation values seem to improve sample quality at the expense of diversity. For CAS, the correlation coefficient between Top-1 Accuracy and FID is 0.16, and IS -0.86, the latter result incorrectly suggesting that improved IS is highly correlated with poorer performance. Moreover, the best-performing truncation values (1.5 and 2.0) have rather poor ISs and FIDs. That these poor IS/FID also seem to indicate poor performance on this metric is no surprise; that other models, with well-performing ISs and FIDs yield poor performance on CAS suggests that alternative metrics are needed. One can easily diagnose the issue with IS: as noted in [28], IS does not account for intra-class diversity, and a training set with little intra-class diversity may make the classifier fail to generalize to a more diverse test set.

Table 2: CAS for VQ-VAE-2 model reconstructions and BigGAN-deep models at different truncation levels at 256×256 resolution.

| Training Set | Truncation | Top-5 Accuracy | Top-1 Accuracy | IS | FID-50K |
|---|---|---|---|---|---|
| BigGAN-deep | 0.20 | 13.24% | 5.11% | **339.06 ± 3.14** | 20.75 |
| BigGAN-deep | 0.42 | 28.68% | 13.30% | 324.62 ± 3.29 | 15.93 |
| BigGAN-deep | 0.50 | 32.88% | 15.66% | 316.31 ± 3.70 | 14.37 |
| BigGAN-deep | 0.60 | 45.01% | 25.51% | 299.51 ± 3.20 | 12.41 |
| BigGAN-deep | 0.80 | 56.68% | 32.88% | 258.72 ± 2.86 | 9.24 |
| BigGAN-deep | 1.00 | 62.97% | 39.07% | 214.64 ± 2.01 | **7.42** |
| BigGAN-deep | 1.50 | 65.92% | 42.65% | 109.39 ± 1.56 | 11.78 |
| BigGAN-deep | 2.00 | 64.37% | 40.98% | 49.54 ± 0.98 | 28.67 |
| VQ-VAE-2 reconstructions | - | **89.46%** | **69.90%** | 203.89 ± 2.55 | 8.69 |
| Real | - | 91.47% | 73.09% | 331.83 ± 5.00 | 2.47 |

FID should better account for this lack of diversity at least grossly, as the metric, calculated as $FID(P_x, P_y) = \|\mu_x - \mu_y\|^2 + tr(\Sigma_x + \Sigma_y - 2(\Sigma_x \Sigma_y)^{1/2})$, compares the covariance matrices of the data and model distribution. By comparison, CAS offers a finer measure of model performance, as it provides us a per-class metric to identify which classes have better or worse performance. While in theory one could calculate a per-class FID, FID is known to suffer from high bias [24] for a low number of samples, suggesting that per-class estimates would be unreliable.[4]

Perhaps a larger issue is that IS and FID heavily penalize non-GAN models, suggesting that these heuristics are not suitable for inter-model-class comparisons. A particularly egregious failure is that IS and FID aggressively penalize certain types of samples that look nearly identical to the original dataset. For example, we computed CAS, IS, and FID on the ImageNet training set at 256×256 resolution and on reconstructions from VQ-VAE-2. As shown in Figure 4, the reconstructions look nearly identical to the original data. As noted in Table 2, however, IS decreases by 128 points and FID increases by 3.5×. The drop in performance is so great that BigGAN-deep at 1.00 truncation achieves better IS and FID than nearly-identical reconstructions. CAS Top-1 and Top-5 for the reconstructions, however, drops by 2.2% and 4.4%, respectively, relative to the original dataset. CAS for BigGAN-deep model at 1.00 truncation, on the other hand, drops by 31.1% and 46.5% relative.

## 4.4 Naive Augmentation Score

To calculate NAS, we add to the original ImageNet training set 25%, 50%, or 100% more data from each of our models. The original ImageNet training set achieves a Top-5 accuracy of 92.97%.

Although the CAS results for BigGAN-deep, and to a lesser extent VQ-VAE-2, suggest that augmenting the original training set with model samples will not result in improved classification performance, we wanted to study whether the relative ordering on the CAS experiments would hold for the NAS ones. Figure 5 illustrates the performance of the classifiers as we increase the amount of synthetic training data. Perhaps somewhat surprisingly, BigGAN-deep models that sample from lower truncation values, and have lower sample diversity, are able to perform better for data augmentation compared to those models that performed well on CAS. In fact, for some of the lowest truncation values, we found a modest improvement in classification performance: roughly 0.2%. Moreover, VQ-VAE-2 underperforms relative to BigGAN-deep models. Of course, the caveat is that the former model does not yet have a mechanism to trade off sample quality from sample diversity.

The results on augmentation highlight different desiderata for samples that are added to the dataset rather than replaced. Clearly, the samples added should be sufficiently different from the data to allow the classifier to better generalize, yet poorer sample quality may lead to poorer generalization compared to the original dataset. This may be the reason why extending the dataset with samples generated from a lower truncation value —which are higher-quality, but less diverse—perform better

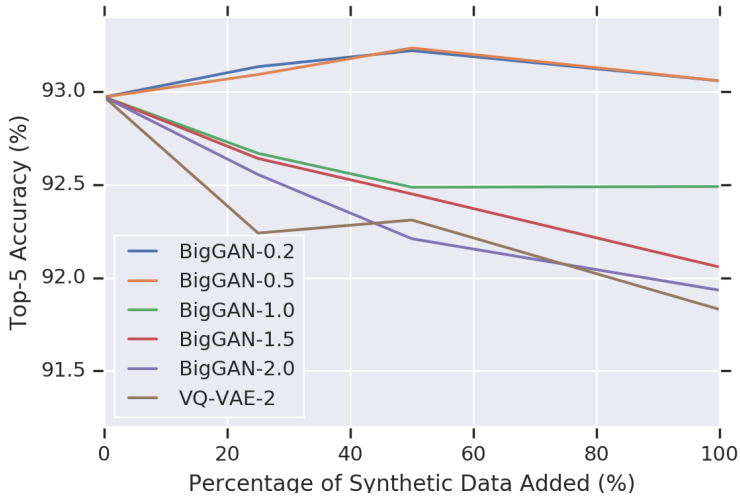

Figure 5: Top-5 accuracy as training data is augmented by $x\%$ examples from BigGAN-deep for different truncation levels. Lower truncation generates datasets with less sample diversity.

Table 3: CAS for different models of CIFAR-10. PixelCNN-Bayes denotes classification accuracy using exact inference using the generative model.

|          | Real   | BigGAN | cGAN   | PixelCNN | PixelCNN-Bayes | PixelIQN |
|----------|--------|--------|--------|----------|----------------|----------|
| Accuracy | 92.58% | 71.87% | 76.35% | 64.02%   | 60.05%         | 74.26%   |

on NAS than CAS. Furthermore, this may also explain why IS, FID, and CAS are not predictive of NAS.

## 4.5 Model Comparison on CIFAR-10

Finally, we also compare CAS for different model classes on CIFAR-10. We compare four models: BigGAN, cGAN with Projection Discriminator [30], PixelCNN [19], and PixelIQN [40]. We train a ResNet-56 following the training procedure of [37]. More details can be found in Appendix A.2. Similar to the ImageNet experiments, we find that both GANs produce samples with a certain degree of generalization. GANs also significantly outperform PixelCNN on this benchmark. Furthermore, since PixelCNN is an exact likelihood model, we can compare classification performance with exact inference using Bayes rule to that with approximate inference using a classifier. Perhaps surprisingly, CAS for PixelCNN is modestly better than classification accuracy using exact inference, though both results are similar. Finally, PixelIQN has similar performance to the newer GANs.

## 5 Conclusion

Good metrics have long been an important, and perhaps underappreciated, component in driving improvements in models. It may be particularly important now, as generative models have reached a maturity that they may be deployed in downstream tasks. We proposed one, *Classification Accuracy Score*, for conditional generative models of images and found the metric practically useful in uncovering model deficiencies. Furthermore, we find that GAN models of ImageNet, despite high sample quality, tend to underperform models based on likelihood. Finally, we find that IS and FID unfairly penalize non-GAN models.

An open question in this work is understanding to what extent these models generalize beyond the training set. While current results suggest that even state-of-the-art models currently underfit, recent progress indicates that underfitting may be a temporary issue. Measuring generalization will then be of primary importance, especially if models are deployed on downstream tasks.

## Acknowledgments

We would like to thank Ali Razavi, Aaron van den Oord, Andy Brock, Jean-Baptiste Alayrac, Jeffrey De Fauw, Sander Dieleman, Jeff Donahue, Karen Simonyan, Takeru Miyato, and Georg Ostrovski for discussion and help with models. Furthermore, we would like to thank those who contacted us, pointing us to prior work.

## Footnotes

[2] It is more correct to state that the losses yield errors, but we present results as accuracies instead as they are standard in computer vision literature.

[3] N.B. IS and FID also suffer the same failure mode.

[4][24] proposed KID, an unbiased alternative to FID, but the variance of this metric is too large to be reliable when using the number of per-class samples in the ImageNet training set (roughly 1,000 per class), and is worse when using the 50 in the validation set. In addition to suffering high bias, per-class FID requires estimation of the real data covariance matrix of rank 2048 using far fewer samples, leading to rank deficiency.

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
