[Supplementary Material · CAS_NeurIPS2019_camera_ready_supp.pdf]

# Appendix A    Experimental Setup

## A.1    ImageNet

We use a ResNet-50 classifier with single-crop evaluation for our models on ImageNet. The classifier is trained for 90 epochs using TensorFlow's momentum optimizer with a learning rate schedule linearly increasing from 0.0 to 0.4 for the first 5 epochs and decreasing by a factor of 10 at epochs 30, 60, and 80. It mirrors the 8,192 batch setup of [39] with gradual warmup. These were trained on 128 TPU chips, and training completed in roughly 45 minutes. We also compared results to those trained on 8 TPU chips with batch size 1,024 (and completed in roughly 10 hours), and found that Top-1 and Top-5 accuracy were within 0.4%.

For BigGAN-deep models, since the truncation trick – which resamples dimensions that are outside the mean of the distribution – seems to trade off quality for diversity, we perform experiments for a sweep of truncation parameters: 0.2, 0.42, 0.5, 0.8, 1.0, 1.5, and 2.0.[5]

## A.2    CIFAR-10

We use a ResNet-56 classifier for our models on CIFAR-10, using 45,000 samples for the training set and 5,000 for validation. We train for 182 epochs, starting at learning rate 1.0 and decaying by a factor of 10 at epochs 91 and 136. We use batch size 128. Note that this setup mirrors the one in [37].

# Appendix B    Instructions for Operating on Google Cloud

(these broadly follow `https://cloud.google.com/tpu/docs/tutorials/resnet`, with some changes for the metric)

For first-time users: 1. Create a project using: `https://console.cloud.google.com/cloud-resource-manager`

2. Enable Billing: `https://cloud.google.com/billing/docs/how-to/modify-project`

3. Create a storage bucket (used for storing data and models): `https://console.cloud.google.com/storage/browser`. Make sure this is in the zone us-central, as this has the cheapest pricing.

For 10-hour training:

1. Launch google cloud shell (`https://cloud.google.com/shell/`)

2.

```
  ctpu up --machine-type n1-standard-8
--tpu-size=v2-8 --preemptible --zone us-central-<x>
```

(where <x> is a,b for paying customers, and f for those in the TFRC program)

3. You will now be in a virtual machine. Run tmux to keep a persistent ssh.

4. run

```
  export PYTHONPATH="$PYTHONPATH:/usr/share/tpu/models"
```

5.

```
  cd /usr/share/tpu/models/official/resnet/
```

6. Set

```
TRAIN_DIR=gs://<BUCKET-NAME>/<synthetic data>
```

to TFRecords of your synthetic data.

7. Set

```
EVAL_DIR=gs://<BUCKET-NAME>/<real data>
```

to TFRecords of the validation data

8. Set

```
MODEL_DIR=gs://<BUCKET-NAME>/<MODEL_DIR>
```

9.

```
python resnet_main.py --tpu=${TPU_NAME} --data_dir=${TRAIN_DIR} \
  --model_dir=$MODEL_DIR \
  --hparams_file=configs/cloud/v2-8.yaml \
  --mode=train
```

10.

```
python resnet_main.py --tpu=${TPU_NAME} --data_dir=${EVAL_DIR} \
  --model_dir=$MODEL_DIR \
  --hparams_file=configs/cloud/v2-8.yaml \
  --mode=eval
```

11. exit shell

12.

```
ctpu delete --zone <ZONE>
```

to turn off the tpu

For 45-minute training:

1. Launch google cloud shell (`https://cloud.google.com/shell/`)

2.

```
ctpu up --machine-type n1-standard-8
--tpu-size=v2-128 --preemptible --zone us-central-<x>
```

(where <x> is a,b for paying customers, and f for those in the TFRC program)

3. You will now be in a virtual machine. Run tmux to keep a persistent ssh.

4. run

```
export PYTHONPATH="$PYTHONPATH:/usr/share/tpu/models"
```

5.

```
cd /usr/share/tpu/models/official/resnet/
```

6. Set

```
TRAIN_DIR=gs://<BUCKET-NAME>/<synthetic data>
```

to TFRecords of your synthetic data.

7. Set

```
EVAL_DIR=gs://<BUCKET-NAME>/<real data>
```

to TFRecords of the validation data

8. Set

```
MODEL_DIR=gs://<BUCKET-NAME>/<MODEL_DIR>
```

9.

```
python resnet_main.py --tpu=${TPU_NAME} --data_dir=${TRAIN_DIR} \
  --model_dir=$MODEL_DIR \
  --hparams_file=configs/cloud/v2-128.yaml \
  --mode=train
```

11. exit shell

12.

```
ctpu delete --zone <ZONE>
```

to turn off the tpu

13. follow steps of 10-hour training, except for steps 6 and 9.

## Footnotes

[5]Dimensions of the noise vector $z$ whose value are greater outside the range of $-2\tau$ to $2\tau$ ($\tau$ is the truncation parameter) are resampled. Lower values of $\tau$ lead to less diverse datasets.