[Reviews · NeurIPS 2019]

Reviewer 1



The paper touches an important problem of measuring the quality of conditional generative models. The author proposed Classification Accuracy Score -- a metric that is based on a performance of a discriminative model that is trained on samples obtained from the conditional generative model. The paper also discussed pros and cons of the proposed metric. The empirical study shows that a number of sota-level deep generative models fail to match the target distribution. Pros: While the idea has been proposed before in Shmelkov2018, it was not widely used in the field. The current paper points out some limitations of deep generative models as well as limitations currently used metrics, thus the paper delivers a significant contribution. The paper is clearly written, the experiments look thoughtfully designed. Cons: The original work Shmelkov2018 has not been cited. The proposed metric has a range of limitations that are listed in section 3 (e.g., the metric does not penalize memorization). Despite the questionable novelty of the proposed metric, the paper provides a nice and interesting empirical evaluation with sota level generative models, that definitely is of interest to the community. The increasing popularity of the proposed metric also has a significant potential impact. I recommending to accept the paper. Comments: 0) The original work Shmelkov2018 should definitely be cited. 1) The experimental evaluation suggests that IS/FD tend to over-penalize likelihood based generative models as well as over-penalize nearly ideal reconstructions from VAEish model. While the issue is discussed in section 4.3 (lines 270-280), it is still not clear why does this happen. I feel like there are some reasons for it that might be interesting and useful for the community. 2) The paper may also benefit from adding comparison on MNIST dataset as well as more extensive evaluations on CIFARs. The first, it is really interesting if the proposed metric is already perfect for a simpler dataset (for most generative models perhaps)? The second, a lot of groups have no resources to reproduce the ImageNet experiments, so more MNIST/CIFARs experiments might accelerate future research. 3) The one more case when the proposed metric might produce wrong results is Dataset Distillation (https://arxiv.org/pdf/1811.10959.pdf) however the case looks extremely unlikely. 4) Formatting recommendation: it is common practice to use a vector format for plots e.g., use pylab.savefig('name.pdf'). (Shmelkov2018) "How good is my GAN?." Proceedings of the European Conference on Computer Vision (ECCV), 2018.

Reviewer 2



The paper presents a new metric to evaluate generative models by training a classifier using only sampled images. Properly evaluating generative models is an important task and this is an interesting novel idea, however I think this tests more the conditional part of conditional generative models then the generative part and the results should be seen as such. Detailed remarks: 1) Conditional generative models have a hard time capturing the class, for example in " Are generative classifiers more robust to adversarial attacks?" the authors get bad classification on cifar10 using a conditional generative model. This was also discussed in "Conditional Generative Models are not Robust" (concurrent work so no need to cite, but might be of interest). It seems that there is a big difference between generative model and conditional generative model and the metric evaluates the latter and should be described as one. Some discussion on the matter is needed. 2) In the same line, it would be important to how the evaluated models capture the class, what is the accuracy of using p(x|y) as a classifier for real data? What is the accuracy of using samples from p(x,y) using a pretrained classifier (trained with real data).

Reviewer 3



The lack of novelty is quite problematic. Essentially the central idea of the paper "regenerate training set using a generative model and train a classifier on this data to compare with the one trained on real data and thus evaluate the generative model" was described in "How good is my GAN?" K. Shmelkov et al. ECCV'18 (which is not mentioned in related work at all). The remaining contributions look pretty weak without the main one. Conditional GANs learn some classes worse than others. FID (IS as well), being a single number, fails to capture this aspect especially for a dataset as rich as ImageNet. Recent likelihood-based generative models actually perform quite well and don't have completely degenerate classes (as their latent space contains reconstructions as well). IS is extremely problematic and does not really account for diversity [23]. Truncation trick inflates IS by directly restricting diversity of BigGAN output and thus degrades CAS. I'd say we already know all this or at least it is unsurprising. The writing looks pretty good though, the paper is well-structured and easy to read. The main idea is clearly explained, easy to reproduce. But not novel, unfortunately.

[Author Response · NeurIPS 2019]

**Thank you for taking the time to review our paper. We appreciate the effort and consideration.** Before addressing reviewers' specific points, we would like to highlight some previous work two reviewers mentioned, and other work we had found between submission and rebuttal. As the reviewers noted, Shmelkov's "How Good is My GAN?" proposes a very similar metric to CAS. Interestingly, in the context of GANs, the metric has been independently proposed three times before that: as "Adversarial Acc." in "LR-GAN: Layered Recursive Generative Adversarial Networks for Image Generation", as "Train on Synthetic, Test on Real" in "Real-valued (medical) time series generation with recurrent conditional gans", and in "A Classification-Based Study of Covariate Shift in GAN Distributions". None of these previous papers cite each other, but we aren't surprised at the previous proposals as the metric is **very** simple. (These works will be cited on revision.) We view our primary contribution, however, to be a discussion of underappreciated aspects of generative model evaluation and use one, CAS, to measure properties of generative models not captured by FID/IS. We perform an extensive empirical evaluation on SOTA models, particularly those whose IS/FID approach those of the data distribution to 1) demonstrate that strong FID/IS does not imply good performance on inference; 2) highlight specific deficiencies of a generative model (such as dropped modes); and 3) demonstrate that likelihood models are overly penalized by IS/FID. We further open-source the metric, which makes reproducibility easy, fast ($\sim$ 45min.), and practical. Also, in contrast to Shmelkov's paper, we think of the metric not as approximate recall, but as approx. inference. One issue with the approx. recall perspective is that for complex datasets such as ImageNet, one will not obtain 100% "recall", even if training on the true distribution. Treating classification as approx. inference, however, allows us to reason about cases in which acc. is $<100\%$.[1] Secondarily, our conclusions are different: we find that there are likelihood models competitive with GANs, but suffer from poor IS/FID (that paper only uses PixelCNN++).

**Reviewer 1**: **Re: 1)** Over-penalization of likelihood: This topic is fascinating, but we didn't share our thoughts due to a lack of a "smoking gun" experiment. Our view on this issue is that GANs tend to perform better on IS/FID because the inductive biases of the discriminator are similar to that of the convnets used for IS/FID. In essence, the GAN generator learns to mimic convnet level features similar to that of the data, whereas the likelihood for a number of likelihood models are fairly dissimilar to the pool3 layer of an inception network (for example). This will be added to the discussion. **Re: 2)** More extensive CIFAR/MNIST comparisons: On CIFAR10, since the vast majority of conditional models we found were GANs, and we wanted to include other model classes, we tried to include exemplars from each class. We can definitely include 20+ models, including better flow models. For MNIST, we did not include models because most are unconditional, and that MNIST was "solved". You do make a cogent argument that the ease of the dataset is also interesting, and is something we can include in a possible camera-ready. **Re: 3)** Dataset Distillation. Interesting paper! Happy to try it; **Re: 4)** vector graphics: Good point. We'll fix that.

**Reviewer 2**: We broadly agree, but perhaps the paper was presented in a way that fostered disagreement. To hopefully clear up any confusion. **Re:** "tests more the conditional part of conditional generative models". We agree. We tried to make this point in lines 81-90 (a gen. model for speech synthesis has different desiderata than for speech recognition). IS/FID implicitly tests for the former, while CAS explicitly tests for the latter. This is especially useful for downstream tasks. **Re: 1)** Thank you for the pointer to "Conditional Generative Models are not Robust". Interesting work! But it focuses on robust classification of max. lik. models and the results may not extend to non-ML based models. Re: "Are generative classifiers more robust to adversarial attacks?", the authors use a PixelCNN++ on CIFAR10, and likely better (future) gen. models will improve classification performance. Re: that "the metric evaluates [conditional generative models] and should be described as one", we tried to be explicit: it's stated in the title (Classification Accuracy Score for Conditional Generative Models), in the abstract, and first description of the metric (lines 52-60). We can make this more explicit, however. **Re: 2)** We found classifying model samples using a classifier trained on real data does not reveal new information as much of that score is already captured in IS in particular, which calculates statistics on classifiers trained on real data. For completeness, however ($x$ in BigGAN-$x$ is the truncation parameter):

| Acc | BigGAN-0.2 | BigGAN-0.42 | BigGAN-0.5 | BigGAN-0.6 | BigGAN-0.8 | BigGAN-1.0 | BigGAN-1.5 | BigGAN-2.0 | VQ-VAE |
|---|---|---|---|---|---|---|---|---|---|
| Top-5 | 99.47% | 99.32% | 99.21% | 98.87% | 97.78% | 95.45% | 82.74% | 60.19% | 64.65% |

**Reviewer 3**: First, we hope that we didn't give the impression that CAS is superior to IS/FID. Our view is presented in lines 72-75 (FID/IS is orthogonal to CAS). As mentioned above, the metric itself is not novel, but it leads to non-trivial conclusions. In addition to the contributions mentioned above: the over-penalization of VQVAE reconstructions is a practical example where reliance on FID leads to incorrect generative model ordering. Second, we included the section of BigGAN truncation parameter sweeps to illustrate that CAS captures different properties than IS/FID. **Re: Prec./Recall** Due to space, let's consider the NVIDIA P/R on ImageNet. The class-specific recall only identifies within-class variation, and any CAS comparison seemed unfair. Second, P/R of the true distribution is far from optimal: unconditional P/R for the validation set, is .67/.66, far less than 1.0/1.0, and misses much of the variation data as there are only 50 egs/class in the validation set. An unfair comparison would say since CAS uses features trained on synthetic images, while P/R uses VGG features trained on real data, the former could highlight model biases. If a model only produces "two-headed" frogs, the CAS classifier might have a "two-head" discriminative feature, but the P/R VGG features will not. But it seems a disservice to the P/R authors, since P/R measures different things well.

## Footnotes

[1] To test the validity of the approximate inference approach, we compared classification accuracy on Cifar10 using exact inference with a PixelCNN (60.05%) vs. CAS (64.02%).


[Meta-Review · NeurIPS 2019]

The paper discusses a classification accuracy score for evaluating conditional generative models. The final version needs to be significantly revised to account for closely related work such as Shmelkov (2018). The novelty of the proposed metric is questionable and should not be misleading in the text. On the other hand, reviewers were impressed with the empirical evaluation, and felt that the paper would provide new insights to the NeurIPS community.